Evidence of acclimatization or adaptation in Hawaiian corals to higher ocean temperatures

Coles Steve L. 1 2
Bahr Keisha D. kbahr@hawaii.edu 2
Rodgers Ku’ulei S. 2
May Stacie L. 2
McGowan Ashley E. 2
Tsang Anita 2
Bumgarner Josh 2
Han Ji Hoon 2
1 Bernice Pauahi Bishop Museum, Department of Natural Sciences , Honolulu , Hawai’i , USA
2 University of Hawai’i at Mānoa, Hawai’i Institute of Marine Biology , Kāne’ohe , Hawai’i , USA
Ries Leslie
Electronic publication date: 2018 Aug 7
Publication date: 2018
Volume: 6
Electronic Location ID: e5347
Received 2018 Feb 17; Accepted 2018 Jul 9
Copyright: ©2018 Coles et al.
Copyright year: 2018
Copyright holder: Coles et al.
License: This is an open access article distributed under the terms of the Creative Commons Attribution License, which permits unrestricted use, distribution, reproduction and adaptation in any medium and for any purpose provided that it is properly attributed. For attribution, the original author(s), title, publication source (PeerJ) and either DOI or URL of the article must be cited.
License URL: https://creativecommons.org/licenses/by/4.0/

Keywords: Climate change, Coral reefs, Adaptation, Bleaching, Ocean warming, Acclimatization

Funding: National Institute of General Medical Sciences, National Institutes of Health P20GM103466 This research was partially funded with support from the National Institute of General Medical Sciences, National Institutes of Health, award number: P20GM103466. There was no additional external funding received for this study. The funders had no role in study design, data collection and analysis, decision to publish, or preparation of the manuscript.

==============================
Ocean temperatures have been accelerating at an alarming rate mainly due to anthropogenic fossil fuel emissions. This has led to an increase in the severity and duration of coral bleaching events. Predicted projections for the state of reefs do not take into account the rates of adaptation or acclimatization of corals as these have not as yet been fully documented. To determine any possible changes in thermal tolerances, manipulative experiments were conducted to precisely replicate the initial, pivotal research defining threshold temperatures of corals nearly five decades ago. Statistically higher calcification rates, survivorship, and lower mortality were observed in Montipora capitata, Pocillopora damicornis, and Lobactis scutaria in the present study at 31 °C compared to the original 1970 findings. First whole colony mortality was also observed to occur sooner in 1970 than in 2017 in M. capitata (3 d vs. 15 d respectively), L. scutaria (3 d vs. 17 d), and in P. damicornis (3 d vs. 13 d). Additionally, bleaching occurred sooner in 1970 compared to the 2017 experiment across species. Irradiance was an important factor during the recovery period for mortality but did not significantly alter calcification. Mortality was decreased by 17% with a 50% reduction in irradiance during the recovery period. These findings provide the first evidence of coral acclimatization or adaptation to increasing ocean temperatures for corals collected from the same location and using close replication of the experiment conducted nearly 50 years earlier. An important factor in this increased resistance to elevated temperature may be related to removal of the discharge of treated sewage into Kāne‘ohe Bay and resulting decrease in nitrification and eutrophication. However, this level of increased temperature tolerance may not be occurring rapidly enough to escape the projected increased intensity of bleaching events, as evidenced by the recent 2014 and 2015 high coral mortality in Hawai‘i (34%) and in the tropics worldwide.

Introduction

Coral reef ecosystems have high biological value and are critical to the health and livelihoods of human communities throughout the tropical oceans. Commonly called the ‘rainforests of the sea’, coral reefs occupy 0.17% of the world’s ocean area (Smith, 1978), yet support more than 33% of all marine species (Fisher et al., 2015). The increasing frequency and severity of coral bleaching events is placing these critical habitats under imminent threat, with some models projecting the collapse of reefs worldwide once climate change exceeds 1–2 °C above the preindustrial ocean temperatures, a value certain to be exceeded by the end of the century (Hoegh-Guldberg, 1999; Frieler et al., 2013).

Coral bleaching is a term first used by Vaughan (1911) to describe the results of low tide exposure of corals at the Tortugas. However, what Vaughan observed was not coral bleaching but mortality and exposure of bare coral skeleton. Yonge & Nicholls (1931) described coral bleaching following temperature increases up to 37 °C in coral pools at the Low Isles, Great Barrier Reef that resulted in the loss of the intracellular symbiotic algae zooxanthellae. If this symbiosis is disrupted for extended periods, pigment loss and eventual mortality will occur (Jokiel & Coles, 1990; Williams & Bunkely-Williams, 1990; Glynn, 1991; Glynn, 1993; Brown, 1997; Wilkinson et al., 1999; Boesch, Field & Scavia, 2000; Fitt et al., 2001; Coles & Brown, 2003; Hughes et al., 2003; Hughes et al., 2017; Hoegh-Guldberg et al., 2007). The first recent documented widespread bleaching event occurred in 1983 off Panama (Glynn, 1991; Graham, 1994), followed by frequent and severe events around the world in 1998 and 2010, (http://www.noaa.gov/media-release/noaa-declares-third-ever-global-coral-bleaching-event). The most recent worldwide bleaching event lasted from 2014 to 2017, the longest, most widespread, and possibly the most damaging coral bleaching on record (Hughes et al., 2017). More than 70% of coral reefs around the world experienced heat stress related to bleaching and/or mortality during the three-year global event (https://coralreefwatch.noaa.gov/satellite/analyses_guidance/global_coral_bleaching 2014-17_status.php, Eakin et al., 2016).

Average offshore Sea Surface Temperatures (SSTs) in Hawai‘i have steadily increased a significant 1.13 °C over the last 47 years (1958–2014) (Fig. 1). The first documented widespread bleaching event in the main Hawaiian Islands occurred in 1996 in Kāne‘ohe Bay (Jokiel & Brown, 2004), followed by major events in 2002 and 2004 in the Northwestern Hawaiian Islands (Aeby et al., 2003; Jokiel & Brown, 2004; Kenyon et al., 2006) and minor bleaching events in 2007 and 2009. These bleaching events were short in duration, and coral communities recovered rapidly once temperatures returned to normal. However, in 2014 and 2015, the Hawaiian Archipelago experienced unprecedented widespread bleaching and mortality with well over half of the corals exhibiting bleaching and a mean mortality of 34% (Kramer et al., 2016; SSR Institute, 2017; Couch et al., 2017; Coles, 2017; Bahr, Rodgers & Jokiel, 2017; Rodgers et al., 2017).

Figure 1 Offshore sea surface temperatures in Hawai‘i.

Sea surface temperatures (SST) for offshore Hawai‘i using IGOSS-NMC temperature data (1992–2014) and corrected NMFS data (1956–1992) for Koko Head, O‘ahu between 1956- 2017.

In the 1970s, threshold temperatures initiating bleaching of Hawaiian corals were determined in a series of pioneering studies using a manipulative flow through seawater system and field observations (Coles, 1973; Jokiel & Coles, 1974; Jokiel & Coles, 1977; Coles & Jokiel, 1977; Coles & Jokiel, 1978). This research compared thermal tolerances for Kāne‘ohe Bay corals and Eniwetok coral congeners and conspecifics and found similar thresholds of +1–2 °C above summer ambient to induce bleaching, indicating corals worldwide to be living within 1–2 °C of their upper limit during summer months. This 1–2 °C bleaching threshold has since been observed across the geographic range that corals inhabit, even though maximum seasonal temperature among regions varies by up to 9 °C, from 25 °C at Rapa Nui to 33–34 °C in the Arabian Gulf (Coles & Brown, 2003; Jokiel, 2004; Jokiel & Brown, 2004; Coles & Riegl, 2013). This seminal research on coral temperature thresholds has been and continues to be used in numerous management applications and modeling projections, including the primary NOAA Coral Reef Watch satellite monitoring and modeled outlook program (Liu et al., 2014). Since both the magnitude of thermal stress and the duration are key factors in bleaching response, the Coral Reef Watch has developed a thermal stress index, the Coral Bleaching Degree Heating Week (DHW), that takes both of these factors into account.

Irradiance is another key factor that determines the extent of bleaching (Coles, 1973; Jokiel & Coles, 1977; Coles & Jokiel, 1978; Brown et al., 2000) and is often correlated with high temperatures which occur during the summer when irradiance is at its maximum (Hoegh-Guldberg & Smith, 1989; Goenaga & Canals, 1990; Fitt & Warner, 1995; Brown et al., 1999; Jokiel, 2004). Individual corals typically show more pronounced bleaching and mortality on upper surfaces of the colony exposed to higher irradiance (Jokiel & Coles, 1977) as do shallower corals exposed to higher irradiance than their deeper counterparts (Jokiel, 2004).

Projections of future climate change have led many to conclude that globally coral reefs could collapse within the next few decades (Hoegh-Guldberg, 1999; Hoegh-Guldberg et al., 2007; Veron et al., 2009; Frieler et al., 2013). The upper thermal tolerance thresholds of corals throughout the world could be exceeded every summer by the year 2030 (Hoegh-Guldberg, 1999). However, many of these projections do not consider the potential for acclimation, acclimatization, or adaptation (Coles & Brown, 2003; Berkelmans & van Oppen, 2006). Acclimation refers to changes in tolerances under laboratory or other experimental conditions, generally over the short-term (Coles & Brown, 2003). Acclimatization refers to phenotypic changes by an organism to stresses in the natural environment that result in the readjustment of an individual organism’s tolerance levels within the organism’s lifetime as it adjusts to a change in the environment. Adaptation is a selective process that may take place over one or many generations when the more stenotopic members of a population are eliminated by environmental stress, leaving the more tolerant organisms to reproduce and recruit to available habitat. Many assume that adaptation is inherently a slow process requiring thousands to millions of years, and that corals will show little adaptive response to the rapid climate change occurring today (Brown, 1997; Gates & Edmunds, 1999; Hoegh-Guldberg, 1999; Hoegh-Guldberg et al., 2007; Veron et al., 2009; Frieler et al., 2013). However, rapid genetic adaptation to temperature in Acropora millipora has been predicted through simulations of higher latitude migration of heat tolerant alleles (Matz et al., 2018). In this paper we will refer to a shift in thermal tolerance as acclimatization/adaptation as it is not within the realm of this research to differentiate between them.

While Hughes et al. (2017) and Hughes et al. (2018) found little evidence to support acclimatization/adaptation of corals to thermal stress related coral bleaching from previous events, there have been numerous reports throughout the world of reduced bleaching and mortality by corals exposed to previous bleaching in the Arabian Gulf (Riegl, 2003), Great Barrier Reef (Maynard et al., 2008; Guest et al., 2012), Caribbean (Castillo et al., 2012), Moorea, French Polynesia (Pratchett et al., 2013), Kiribati (Carilli, Donner & Hartmann, 2012) and the Northwestern Hawaiian Islands (Couch et al., 2017). Other studies indicate that acclimatization or adaptation to increased temperature is possible in many coral species. Coral species in similar habitats have shown different bleaching susceptibilities, indicating that some coral species and even individuals within species are more resistant to environmental stressors than others (Baird & Marshall, 2002; Stimson, Sakai & Sembali, 2002; Grottoli, Rodrigues & Juarez, 2004). Also, the adaptation in thermal tolerance corresponding to local ambient temperatures worldwide is an indication of long-term selection for more temperature tolerant corals determined by a region’s thermal history (Jokiel & Coles, 1990; Coles & Brown, 2003). However, rates of acclimatization/adaptation to projected rapid climate change and escalating stress from rising seawater temperatures have not been determined. Lacking specific experimental information, a mathematical model by Donner et al. (2005) has proposed that bleaching will become an annual event in the next 30–50 years without an increase in thermal tolerance of 0.2–1.0 °C per decade. Genomic model simulations by Bay et al. (2017) suggest that corals have sufficient genetic variability for rapid evolution of heat tolerance for survival under mild ocean warming but may undergo extinction under high CO2 emissions. Logan et al. (2014) modeled possible adaptive responses in corals and found that an adaptive increase in bleaching thresholds would delay bleaching occurrence by only 10 additional years.

Kāne‘ohe Bay has some of the highest coral cover in Hawai‘i, but has historically been 1–2 °C higher in the summer months than most other reef areas in Hawai‘i due to restricted circulation (Jokiel & Brown, 2004; Franklin, Jokiel & Donahue, 2013; Bahr, Jokiel & Toonen, 2015a). Additionally, the coral reefs of Kāne‘ohe Bay have been surprisingly resilient over decadal timescales following numerous events that drastically reduced live coral cover (Bahr, Rodgers & Jokiel, 2017). The gradual rise in open ocean water temperatures of nearly 1 °C over the last 50 years (Fig. 1), and the occurrence of three major bleaching events in Kāne‘ohe Bay that were followed by high recoveries of the dominant reef building corals provided a unique opportunity to assess whether acclimatization/adaption of these corals has occurred. To assess possible changes in thermal tolerances we replicated experiments conducted in 1970 that first defined the long-term thermal tolerances of corals (Coles, 1973; Jokiel & Coles, 1974; Coles, Jokiel & Lewis, 1976; Jokiel & Coles, 1977; Coles & Jokiel, 1978). Ambient temperature in the experiments in 1970 (26.4 °C) was 2.2 °C lower than in the current 2017 experiments (28.6 °C). We thus exposed corals to temperatures 2.8 °C above ambient to replicate the increase in temperature above ambient in the 1970 experiments. Replication was also controlled through identical methodology, location, seawater system, and observer. Results are evaluated in terms of the long-term potential for upward adjustments of thermal thresholds for coral bleaching and interspecific differences among species for capacity to change.

Methodology

The experiments conducted in 1970 and 2017 were conducted in the identical fiberglass flow-through mesocosms exposed to full sunlight at the same coral reef ecology laboratory. This open flow experimental system assures a representative environment to track biological response under natural conditions (Jokiel, Bahr & Rodgers, 2014). Corals were collected from shallow Kāne‘ohe Bay reef flats (collection permit DAR SAP 2018-03) at depths approximately equal to the 0.5 m deep mesocosms, therefore the unshaded light intensity in the mesocosms are nearly identical to the light corals experience in the field (Jokiel & Coles, 1977). In our 2017 experiment, we examined the interaction between irradiance and temperature (ambient temperature, +2.8 °C above ambient), with full natural light intensity and shading (50%) for both temperature treatments. Six 1 m × 1 m mesocosms (n = 3 per temperature treatment), and a water delivery system with flow rates identical to the ones designed by the original authors were used in this experiment. The seawater intake is located 20 m from the experimental site at a depth of 2 m.

To replicate the original thermal tolerance experiments conducted in 1970 (Coles, Jokiel & Lewis, 1976; Jokiel & Coles, 1977; Coles & Jokiel, 1978), three species of corals, Montipora capitata (previously M. verrucosa), Pocillopora damicornis, and Lobactis scutaria (previously Fungia scutaria), (Gittenberger, Reijnen & Hoeksema, 2011) were collected from the Moku o Lo‘e reef in Kāne‘ohe Bay, O‘ahu, near the original 1970 experiment collection site (21.4°N, 157.8°W). The endemic species, Porites compressa, was added in 2017 to compare thermal tolerance of the most abundant coral in the bay but had not been tested in the 1970 experiment due to parasitism of Phestilla sibogae that was uncontrollable at that time. These species represent the diversity of common Hawaiian corals. P. compressa is a gonochoristic broadcast spawner and shows traits associated with both stress-tolerant and competitive life history strategies (Hunter, 1998). M. capitata is a hermaphroditic broadcast spawner and shows a competitive life history strategy (Kolinski, 2004; Padilla-Gamiño et al., 2013). P. damicornis engages in brooding as well as broadcast spawning and shows a weedy life history strategy but reaches a determinate size (Richmond & Jokiel, 1984; Richmond & Hunter, 1990). Unlike these three dominant reef-builders, Lobactis scutaria is a solitary ahermatypic coral that can be abundant on Kāne‘ohe Bay reefs (Darling et al., 2012). During the first bleaching event observed in Kāne‘ohe Bay in 1996 and later events in 2014 and 2015, L. scutaria was one of the most resistant to bleaching; P. compressa and M. capitata showed moderate resistance to bleaching, and P. damicornis showed high levels of susceptibility to bleaching (Jokiel & Brown, 2004; Bahr, Jokiel & Rodgers, 2015b).

Experimental conditions

Twenty colonies of each species were placed in each of the six aerated 660-liter mesocosms, for a total of 480 colonies, and weighed using the buoyant weighing technique (Jokiel, Maragos & Franzisket, 1978). All colonies were placed on white styrene lighting panels and elevated 5 cm off the bottom to reduce sediment collection on corals and facilitate fish cleaning. Each individual coral was tagged using a DYMO labeler and attached with plastic coated wire. To avoid damage to the solitary L. scutaria labels were affixed to the bottom of each coral using ZSPAR A-788 non-volatile Splash Zone compound. Corals were divided into two groups for shading, randomly placed within each section and mesocosm, and allowed to acclimate to conditions for a period of two weeks at ambient mean temperatures of 28.6 °C. Half of each tank was covered with a frame of 50% shade cloth to simulate depth (∼2.5 m) while the other half remained uncovered in full sunlight. Shading was evaluated using a LiCor LI-250A light meter.

To control settlement and growth of the nudibranch Phestilla sibogae, which feeds on Porites compressa, and the flatworm Prostiostomum montiporae, that preys on Montipora capitata, one adult Chaetodon auriga (IACUC permit # 2620) was placed in each of the six mesocosms. To reduce algal growth, one adult Acanthurus triostegus was placed in each mesocosm. Daily addition of frozen brine shrimp supplemented fish feedings. To supplement parasitism control, twice weekly manual cleaning of Phestilla eggs and adults was conducted.

Onset™ Pro v2 temperature loggers with an accuracy of ± 0.21 °C from 0°to 50°C and a drift of 1% yr−1 were calibrated at 0 °C and 35 °C prior to placement in each mesocosm. Loggers were set to record at 15-minute intervals throughout the duration of the experiment. Flow rates were calibrated between mesocosms for a turnover rate of approximately one hour and reassessed weekly throughout the eight-week experiment. Following the acclimation period, three replicate mesocosms remained under ambient conditions. One 800-watt Finnex titanium heater and two 1,000-watt Blue Line IPX8 titanium heaters were positioned in each of the remaining three mesocosms to assure even heating to 31.4 °C, an increase of 2.8 °C above ambient conditions for the 31-day experimental period (11 Jul–11 Aug 2017). AquaClear powerhead multifunctional water pumps and rapid turnover rates assured consistency among corals. All heaters were removed prior to the 28-day recovery period (12 Aug–8 Sept 2017).

Environmental variables

Long-term sea surface temperature trends from the Koko Head site were downloaded from Integrated Global Ocean Services System-National Meteorological Center (http://iridl.ldeo.columbia.edu/SOURCES/.IGOSS/.nmc/.Reyn_SmithOIv2/.weekly/.sst/).

The HIMB meteorological station located on site provided UV, PAR, total solar, wind direction and speed, air and water temperatures, and precipitation data (available at http://www.pacioos.hawaii.edu/weather/obs-mokuoloe/). Salinity, temperature, and dissolved oxygen were measured daily in each mesocosm using a YSI 556 MPS multimeter (Table 1). Historical environment parameters were compiled from previous studies (Smith et al., 1981; Cox, Ribes & Kinzie III, 2006) for comparison. These values were derived from previous studies with temporal and spatial measurements closest to the 1970 experiment and prior to the sewage diversion in 1977 through 1979 (Table 1). Mesocosm ambient temperatures were 2.2 °C higher in 2017 than in 1970 during the same July–August periods. Salinity (ppt), PAR (k), and Total N&P (µMol/L) not measured in the mesocosms in 1970 were obtained from measurements taken in 1976-1977 in the adjacent South Bay (Smith et al., 1981) and compared to values from 1998-2001 (Cox, Ribes & Kinzie III, 2006) from the same site for pre- and post-sewage values.

Table 1 2017 experimental conditions.

Environmental characteristics during the 31 d stress period (11 July–11 Aug 2017).

Treatment	Tank number	Mid-day temperature	Salinity	DO (mg L−1)	Irradiance (µmol photons m−2 s−1)	
		Mean	± SE	Mean	± SE	Mean	± SE	Mean	± SE	Max	
Ambient	5	28.46	±0.02	34.65	±0.06	10.40	±0.38	749.18	±23.42	1,745.00	
7	28.54	±0.03	34.66	±0.06	10.43	±0.61	
12	28.83	±0.03	34.64	±0.05	10.43	±0.46	
Heated	4	31.46	±0.02	34.67	±0.06	9.46	±0.55	
8	31.40	±0.02	34.68	±0.06	9.53	±0.41	
11	31.35	±0.02	34.68	±0.06	9.30	±0.40	

Calcification

The initial and final buoyant weights were converted to dry skeletal weight for each species (Jokiel, Maragos & Franzisket, 1978). These data were expressed as the mean solid radius, which uses cube-root approximation to compute a one-dimension linear estimate. Therefore, the calcification rate is expressed as a change in length of the radius rather than the weight change. This transformation compensates and permits for comparison of colony calcification independent of corallum sizes and morphology (Maragos, 1978).

Partial Mortality

Partial mortality was defined as percentage of dead skeletal area on each coral colony throughout the 31-day experimental period and subsequent 28-day recovery period. Partial mortality was scored in bins of 10% twice weekly. Values ranged from zero (no mortality) through various amounts of tissue loss (partial mortality) to 100% (whole-colony mortality) (Baird & Marshall, 2002). Survivorship was characterized by the number of individuals alive within a treatment during the experimental and recovery period. These data were used to compare the 2017 with the 1970 results.

Visual assessments

Visual assessments of condition of corals in all treatments were made by author SLC, who collaborated in the 1970s experiment observations or personally conducted these observations (Coles, 1973; Coles, Jokiel & Lewis, 1976; Coles & Jokiel, 1978). During the elevated temperature phase of the 2017 experiment, observations of coral mortality and pigmentation were made in both heated and ambient tanks at approximately the same frequency as recorded by Jokiel & Coles (1977) (i.e., 2–3 times per week during the first two weeks, twice a week during the third week and once a week during the fourth week). Following temperature reduction to ambient in all tanks, corals were observed approximately weekly for another 28 days to determine any recovery that might occur.

The original 1970 experiment used four categories to describe coral condition: Normal, pale, bleached, and dead. The appearance of these has been previously described in Jokiel & Coles (1974). Images of the four stages of pigmentation are available in Fig. S1.

Statistical approach

2017 experiment

Mean mid-day water temperatures within treatment mesocosms were analyzed using a one-way ANOVA during the experimental and recovery periods. Percentage data for partial mortality at the end of the experimental (11 Aug 2017) and recovery phases (8 Sept 2017) were transformed using an arcsine cubed root transformation and subsequently analyzed with a three-way ANOVA model with fixed factors of temperature, irradiance level, and species. Assumptions of normal distribution and homoscedasticity were assessed through graphical analyses of the residuals. An unbalanced design was used to account for an imbalance in sample size between the shaded and unshaded treatments due to a technical error. Type III sums of squares estimated the main effect of the squared differences of the unweighted marginal means.

Coral calcification at the end of the experimental period (Day 31) was analyzed using a General Linear Model (GLM). Corals subjected to high levels of partial mortality (>50% tissue loss) at the end of the 31-day experiment were removed from the GLM calcification analysis. Type III sums of squares were used to estimate the main effect of the squared differences of the unweighted marginal means. Descriptive and statistical analyses were conducted in JMP Pro 12 (SAS Institute Inc. USA).

Comparison of 1970 vs 2017 experiments

The long-term open-ocean Koko Head temperature trends (Integrated Global Ocean Services System-National Meteorological Center (http://iridl.ldeo.columbia.edu/SOURCES/.IGOSS/.nmc/.Reyn_SmithOIv2/.weekly/.sst/) between 1970 and 2017 were analyzed using a linear regression and mean sea surface temperatures with a one-way ANOVA.

Coral survivorship at heated temperatures (31 °C) was analyzed using a cox proportional hazards regression analysis by year, with censoring of individuals that survived to the end of the experiment. Coral time to mortality was recorded as the number of days since the start of the experiment within each year. Wilcoxon rank-sum test was used to compare the average number of day until the onset of bleaching and whole-colony mortality between years within species.

Results

2017 Experiment

Experimental conditions

During the 2017 experimental period (11 Jul–11 Aug 2017), the corals received full natural solar radiation at mid-day maximum irradiance levels of 1,745 µmol photons m−2 s−1 and mean net irradiance fluxes of 749.18 µmol photons m−2 s−1. Within treatments of temperature (mean mid-day temperatures) were not significantly different between mesocosms (One-way ANOVA; F (2,53) = 1.28; p = 0.27). Experimental mesocosms were heated to a mean of 31.40 ± 0.015 °C and ambient conditions were 28.62 ± 0.015 °C (Table 1). Heaters were removed from the mesocosms on 11 Aug 2017 and the recovery period commenced (12 Aug–8 Sept 2017). Temperatures did not differ among mesocosms during the recovery period (Oneway ANOVA; F(5, 185) = 1.80; p = 0.12).

Calcification

Compared to the control treatment (28.6 °C), increased temperatures (31.4 °C) significantly reduced calcification rates (pooled among irradiance; mean difference ± SE) in Montipora capitata (−63%; -3.03 ± 0.26 mm d−1; p < 0.0001), Pocillopora damicornis (−55%; −1.85 ± 0.29 mm d−1; p < 0.0001), Porites compressa (−51%; −2.09 ± 0.31 mm d−1; p < 0.0001), and Lobactis scutaria (−26%; −0.99 ± 0.25 mm d−1; p < 0.0001) (General Mixed Model; F(11,390) = 21.58; p < 0.0001) (Fig. 2). Calcification response did not vary across irradiance levels (p = 0.442). Once the heat stress was removed, the decline of calcification rates continued with reductions of −89% (mean difference; −1.23 ± 0.26 mm d−1) across species (General Mixed Model; F(15,295) = 2.756; p < 0.0001).

Figure 2 2017 Coral calcification response.

Calcification (mm d−1) of tested coral species Lobactis scutaria, Montipora capitata, Pocillopora damicornis, and Porites compressa under ambient (blue) and heated (red) temperatures at the end of the 31-day stress period (11 July–11 Aug 2017). Error bars are SE of mean response per treatment tank (n = 3). Data are pooled across irradiance regimes. * denotes significant statistical difference.

Partial mortality and recovery

Elevated temperatures caused high rates of tissue loss (mean ± SE) leading to significant increases in exposed dead skeleton (pooled among light levels) in Lobactis scutaria (12 ± 0.04%), Pocillopora damicornis (51 ± 0.06%), Montipora capitata (28 ± 0.05%), and Porites compressa (60 ± 0.06%) at the end of the experimental phase (Three-way ANOVA; F(15, 437) = 16.45; p < 0.0001) (Fig. 3).

Figure 3 2017 Coral partial mortality.

Partial mortality of tested coral species Lobactis scutaria (A), Montipora capitata (B), Pocillopora damicornis (C), and Porites compressa (D) under ambient (blue) and heated (red) temperatures in unshaded and shaded (grey panels) regimes during the 30 day stress (red shaded; 11 July–11 Aug 2017) and following recovery period (12 Aug–8 Sept 2017).

The progressive bleaching and mortality during the experimental phase of the study continued into the recovery phase after temperatures were returned to ambient (Fig. 3). At the end the recovery phase partial mortality varied among species. In heated treatments, the lowest partial mortality was observed in L. scutaria (18%), followed by M. capitata (79%), P. compressa (89%), and P. damicornis (93%) (species*temperature, p < 0.0001; pooled across irradiance levels). Irradiance level (shading vs. unshaded) also played a role in partial mortality during the recovery phase (p = 0.016). Unshaded corals had 17% higher mortality than shaded corals (Three-way ANOVA; F(15, 446) = 64.96; p < 0.0001) (Fig. 3).

Comparison of 1970 vs 2017 experiments

Environmental variables

Offshore mean yearly sea surface temperatures (SST) have increased by 1.13 °C over the past 47 years (R2 = 0.06, F(1,4506) = 288.01, p < 0.0001) (Fig. 1). This increase in SST has created a shifting baseline for comparison of the Jokiel and Coles 1970 experiment with the current experiment. Summer (July–August) mean mid-day ambient temperatures between 1970 (26.4 °C) and 2017 (28.6 °C) differed by 2.2 °C (Oneway ANOVA; F(1,21) = 10.66; p = 0.0039) (Table 2). Temperature variability was slightly greater for the 1970 experiments (e.g., see Coles & Jokiel, 1978; Table 1) where standard deviation (SD) in four tanks of one experimental series ranged 0.9–1.2 °C, while SD for the 2017 experiment ranged from 0.36–0.45 °C in ambient tanks and 0.32–0.36 °C in stress temperature tanks (Table 2). These differences in temperature variability are trivial when compared with the mean temperatures of the treatments.

Table 2 1970 vs. 2017 environmental conditions.

Comparison of environmental parameters in south Kāne‘ohe Bay during experimental periods.

Parameter	1970a	2017b	
Temperature (°C)	26.40 ± 0.03	28.70 ± 0.02	
Salinity (ppt)	34.9 ± 0.09	35.1 ± 0.04	
PAR (k)	0.37	0.29 ± 0.01	
Total N (µMol/L)	10.60	7.38 ± 1.45	
Total P (µMol/L)	1.01	0.32 ± 0.07	
Nitrate + Nitrite (µMol/L)	0.38	0.05 ± 0.12	
Phosphate (µMol/L)	0.48	0.08 ± 0.004	
Notes.

a Temperature data taken from 1970 experiments, all other parameters from .

b Temperature data taken from 2017 experiment, all other parameters from (CISNet data). Mean ± SE.

Calcification

During the 1970 experiment, elevated temperatures had a significant impact on calcification rates across species (−99% L. scutaria; −164% M. capitata; −172% P. damicornis) in comparison to ambient temperatures (26.4 °C) (Fig. 4). When compared to replicated conditions in the 2017 experiment, less severe reductions in calcification rates occurred. Elevated temperatures under ambient irradiance conditions reduced calcification by −8% in L. scutaria, −50% in M. capitata, and −16% in P. damicornis (Fig. 4, Table 3).

Figure 4 1970 vs. 2017 coral calcification.

Calcification of tested coral species during 1970 (white circle) and 2017 (black square) across experimental temperatures. Error bars for 2017 data are SE of mean (n = 3).

Survivorship

Coral survivorship at elevated temperature (∼31.4°C) was higher in 2017 across species: L. scutaria (92%; p < 0.0001), M. capitata (83%; p < 0.0001), and P. damicornis (60%; p = 0.0003) (Cox Proportional Hazards Regression Analysis; Fig. 5; Table 3, Table S1). Lower surivorship was observed at 31 °C in 1970 (40% L. scutaria; 0% M. capitata; 5% P. damicornis).

Table 3 1970 vs. 2017 summary comparison.

Percent change of calcification and survivorship between ambient and elevated temperature conditions at the end of the experiment (n = 20 in 1970 for 30 d, n = 60 in 2017 for 31 d).

	Calcification	Survivorship	
Species	1970	2017	1970	2017	
Lobactis scutaria	−99.23%	−7.84%	40%	92%	
Montipora capitata	−164.29%	−50.16%	0%	83%	
Pocillopora damicornis	−172.73%	−16.04%	5%	60%	

Figure 5 1970 vs. 2017 Coral survivorship.

Survivorship (number of individuals; >95% partial mortality) of tested coral species (A. L. scutaria; B. M. capitata; C. P. damicornis) at 2017 summer ambient temperatures (28.7 °C; blue square), and current heated treatments (31.4 °C; red circle) and in 1970 (31 °C; black triangles). No mortality was observed in 1970 summer ambient treatment (26.4 °C; Jokiel & Coles, 1977). Mean survivorship values were used for 2017 and error bars are SE of mean (n = 3).

Additionally, whole-colony mortality was observed sooner, just after 3 days of exposure to 31 °C, in 1970 (i.e., P. damicornis, n = 15; M. capitata, n = 8; and L. scutaria, n = 3) (Jokiel et al., 1975) (Wilcoxon rank-sum test: L. scutaria p<0.0001; M. capitata p < 0.0001; P. damicornis p < 0.0001).

Corals were able to withstand elevated temperatures (31.4 °C) for a longer period of time in the current 2017 experiment. Entire colony death was first observed in P. damicornis (n = 1) after 13 days, in M. capitata (n = 1) after 15 days and in L. scutaria (n = 1) after 17 days (Fig. 5).

Visual assessment

In 1970, onset of bleaching was observed after 3 days of exposure to elevated temperatures (31.0 °C) in P. damicornis (n = 2), M. capitata (n = 8), and bleaching was observed after 5 days in L. scutaria (n = 7). Bleaching was prolonged during the 2017 experiment (Wilcoxon rank-sum test: L. scutaria p = 0.0002; M. capitata p < 0.0001; P. damicornis p = 0.0002). Initial bleaching was observed after 6 days of exposure to 31.4 °C in M. capitata (n = 1) and P. damicornis (n = 1). Full bleaching was observed in L. scutaria (n = 1) after 8 days of elevated temperature.

These differences of progressive patterns of bleaching and mortality are indicated by the visual assessment average scores shown in Fig. 6. All three species showed more rapid bleaching and mortality in 1970 than in 2017, with complete mortality occurring for M. capitata and P. damicornis by 12 days at 31°C in 1970, which did not occur for either species at 31.4 °C by the end of the experimental period.

Figure 6 1970 vs. 2017 Coral visual assessment.

Comparison of visual assessment scores representing mean “health” (1, normal; 2, pale; 3, bleached, 4, dead) in 1970 and 2017 (full sun only) during the 31 d stress period under heated conditions (2017 red circle and 1970 black triangle) and ambient temperatures (2017 blue square and black diamond) in L. scutaria (A.) M. capitata (B.) and P. damicornis (C.). Error bar represent SE of the mean (n = 20). Examples of visual assessment scoring are provided in Fig. S1.

Discussion

Little is known about the potential for corals to acclimatize/adapt to the rapid pace of climate change. This research assessed the potential for a shift in thermal tolerances of Hawaiian corals over the past half century by replicating the experimental design and using the same observer as in the original 1970 experiment. Although acclimatization/adaptation to increasing local ambient temperatures has occurred in corals globally over the long term in different geographic environments (Coles & Brown, 2003), the rate of acclimatization/adaptation has not been previously determined for rapid temperature increases that occur in severe bleaching events. Our experiments are the first to demonstrate thermal acclimatization/adaptation to elevated ocean temperature for corals of the same species and from the same location over the past half century.

Our results show significant differences in coral bleaching, calcification, survivorship, and mortality since 1970 in three species of corals (L. scutaria, M. capitata, P. damicornis). These corals show higher calcification rates at a similar temperature increase in 2017 as compared to 1970. Calcification rates remained impaired under elevated temperatures across species; however, the reductions in 2017 were not as severe as those documented in 1970. When we compared reductions in calcification rates due to elevated temperatures across years, we found that calcification rates were 70-90% higher in 2017 (Table 3). Similarly, mean mortality across species was substantially reduced in 2017 (22%) as compared to 1970 (85%). In 1970, mortality was high after 30 d of exposure to 31 °C across species (Fig. 5). We observed significantly higher survivorship among species after 31 d at 31.4 °C (Table 3). First whole colony mortality was also observed to occur sooner in 1970 than in 2017 in M. capitata (3 d vs. 15 d respectively), L. scutaria (3 d vs. 17 d), and in P. damicornis (3 d vs. 13 d). In 2017, calcification continued to decline during the recovery period suggesting allocation of resources from growth to repair (Henry & Hart, 2005). Unfortunately, no recovery measurements were reported from 1970. Supporting evidence of acclimatization/adaptation was also observed in the qualitative bleaching assessments. Bleaching was reported much sooner in 1970 as compared to 2017 at similar temperatures (Fig. 6). In 1970, onset of bleaching occurred in half the number of days (3 d) than in 2017 (6 d) in P. damicornis and M. capitata and three days sooner in L. scutaria (5 d vs. 8 d respectively).

The absolute temperature increase above ambient in 1970 (from 26.4 °C to 31.0 °C) was 4.6 °C increase above ambient. In 2017 the increase above ambient was 2.8 °C from 28.6 °C to 31.4°C. This is an increase between 1970 and 2017 ambient temperatures of 2.2 °C. To replicate realistic field conditions and test if thermal tolerances of corals have increased since 1970, temperatures were raised to 31.4 °C, a level similar to 1970 levels where significant bleaching and mortality occurred. Our results thus indicate a shift in the temperature threshold tolerance of these corals to a 31-day exposure to 31.4°C. In 1970, no mortality occurred for corals exposed to 29.6 °C, ∼3 °C above the 1970 ambient (Jokiel & Coles, 1977). It is likely that a temperature increase of 4.5°C above the 2017 ambient would have resulted in the same level of bleaching and mortality at 31 °C as in 1970, confirming that there was a shift upward in thermal tolerance that corresponded to the long term ambient temperature history. This corresponds to a shift upward of ∼2.0 °C in thermal tolerances of Enewetak compared to Hawaiian corals that is related to the long-term temperature environments of the two regions (Coles, Jokiel & Lewis, 1976).

Irradiance has been documented to have a significant influence on coral growth, bleaching, and mortality (Jokiel & Coles, 1977; Coles & Jokiel, 1978; Hoegh-Guldberg & Smith, 1989; Goenaga & Canals, 1990; Fitt & Warner, 1995; Brown et al., 1999; Jokiel, 2004). Our investigation of response across irradiance levels determined that irradiance plays a key role in the recovery of corals. Corals exposed to identical temperatures with a 50% reduction in irradiance had a 17% lower mortality rate than those at full light exposure. However, calcification rates did not differ across irradiance levels during either the stress or recovery periods.

Although experimental conditions were carefully replicated, there were uncontrollable environmental variables that were not identical between the experimental years. There has been an increase in offshore SSTs in Hawaiian waters of 1.13°C over the past 47 years (Fig. 1) and the water quality in Kāne‘ohe Bay has improved considerably (Table 2). At the time of the original experiment in 1970, treated sewage discharge into south Kāne‘ohe Bay had been steadily increasing for the previous 20 years (Smith et al., 1981), elevating inorganic nutrients and reducing visibility by increasing plankton reproduction and growth. Coral abundance in south Kāne‘ohe Bay was minimal at that time compared to the present. Following sewage diversion in 1977–78, average total nitrogen and phosphorus in the water column decreased 30–68%, and inorganic nitrate+nitrite and inorganic phosphate decreased 83–86% by 2006 (Table 2).

These elevated nutrient levels may have contributed to the higher levels of bleaching and mortality that occurred in the 1970 experiments compared to 2017. Considerable evidence has been developed during the last decade indicating that inorganic nutrient loading of water in areas with corals plays a significant role in causing bleaching and mortality of corals at lower temperatures than occur in low nutrient environments. Controlled laboratory experiments by Cunning & Baker (2013) and Baker et al. (2018) have shown that increased dissolved inorganic nitrogen results in increased mitotic indices for symbiotic zooxanthellae, increased algal reproduction and growth, and decreasing translocation of carbon to the coral host. The final result of this process is proliferation of the symbiont at lower temperatures than would be the case in a low nutrient environment and a “parasitizing” of the coral host (Baker et al., 2018) of the energy it would otherwise receive, ultimately leading to formation of reactive oxide species that trigger coral bleaching.

This paradigm, first proposed by Woolridge (2009), has also been substantiated by field measurements on the Great Barrier Reef (Woolridge & Done, 2009; Woolridge et al., 2012). Comparing dissolved inorganic nitrogen (DIN) concentrations and coral bleaching thermal thresholds between inshore reefs receiving high levels of DIN from shore runoff with offshore reefs not subject to runoff, they estimated a >50% reduction in DIN to result in a potential 2 °C increase in bleaching temperature threshold (Woolridge et al., 2012). These findings based upon simultaneous observations of higher bleaching thresholds in lower DIN at different locations are remarkably similar to the higher survival and calcification rates for our experiments at the same location in water with lower DIN after nearly 50 years. Woolridge et al. (2012) suggest that low DIN (<1 µM) water can confer ∼2 °C of resistance to coral bleaching compared to DIN rich (>1–10 µM) water. This is confirmed by the nutrient conditions in Kāne‘ohe in the 1970s and recently. The 0.38 µM for nitrate and nitrite determined by Smith et al. (1981) in the south bay pre-removal of treated sewage, shows ammonium levels of 0.77 µM, for a total of 1.15 µMn DIN prior to 1977. Although no data for ammonium are available from Cox, Ribes & Kinzie III (2006), nitrate and nitrite totaled only 0.05 µM (Table 2), and it is very likely that ammonium decreased proportionally. Earlier Smith et al. (1981) found a 34% decrease in ammonium only a few months after cessation of sewage disposal in the south bay.

These findings thus provide a historical basis of support for the importance of nutrient levels in affecting temperature related coral bleaching thresholds, they emphasize the necessity of managing and limiting anthropogenic related sources of nutrification and eutrophication for sewage discharges from point sources and injection wells, and non-point sources from land-based runoff carrying elevated nutrients from fertilizers and animal feedlots. However, there has been little evidence on a large scale in nature that supports the research results of a reduction of nutrients ameliorating bleaching occurrence (e.g., Bruno & Valdivia, 2016). Pristine reefs along with reefs with high nutrients have been heavily impacted by bleaching. Although our results provide evidence of acclimatization/adaptation to increasing ocean temperatures and indicate that this process can be assisted by controlling nutrification, it is problematical whether corals will be able to survive the IPCC projected rapid increase in temperature levels that are well outside the range of coral survival. Most coral species are expected to exceed their upper lethal limits by 2030 (Hoegh-Guldberg et al., 2007; Veron et al., 2009; Frieler et al., 2013). Some species will be eliminated prior to other more tolerant species as we found with the low mortality of L. scutaria in this study and as reported by others (Bahr, Jokiel & Rodgers, 2016). A shift in species composition and coral diversity is predicted to occur as temperatures increase.

Acclimatization/adaptation of 0.2–1.0 °C per decade has been calculated as necessary to avoid annual bleaching events (Donner et al., 2005). Our July–August mesocosm ambient temperatures in 2017 were 2.5 °C higher than in the experiment conducted nearly five decades earlier, a 0.48 °C per decade increase. However, increased bleaching tolerance may not be enough for coral survival, as evidenced by the 2014/15 bleaching event that reduced coral populations in the main Hawaiian Islands by 34% (SSR Institute, 2017). The slow growth and recruitment of many species of corals combined with repetitive bleaching events of increasing severity and duration may lead to a catastrophic collapse (Intergovernmental Panel on Climate Change, 2014). Moreover, an analysis of worldwide bleaching events from 1980 to 2016 (Hughes et al., 2018) has determined that the median return time between pairs of bleaching events has diminished from once every 25–30 years to only six years since the early 1980s, allowing little time for coral community recovery.

Any climate change mitigation scenarios will require the reduction in use of fossil fuels and lower emissions of CO2 and other greenhouse gases. Coupling effective marine management strategies including nutrient control with acclimatization/adaptation may result in a deceleration of devastating effects from bleaching events and could extend the coral thermal tolerance threshold beyond the predicted 2030 timeline (Hoegh-Guldberg et al., 2007; Veron et al., 2009; Frieler et al., 2013).

Conclusions

• Bleaching: Bleaching occurred sooner in all species tested in 1970 as compared to 2017.

• Survivorship: Higher survivorship was observed in 2017 (92% in L. scutaria; 83% in M. capitata; and 60% in P. damicornis) than in the original 1970 experiment (40% in L. scutaria; 0% in M. capitata; 5% in P. damicornis) when corals were exposed to similar upper lethal temperatures (∼31.4 °C).

• Growth: Results of the 2017 study revealed prolonged exposure (31 d) to upper lethal temperatures (∼31.4 °C) suppressed calcification rates by 26–63% across tested species. Comparison of these reductions in calcification rates to the original 1970 experiment revealed even larger declines in calcification (−172.7% in P. damicornis; −164.3% in M. capitata; −99.2% in L. scutaria) (Table 3).

• Recovery: Results of this current study show irradiance to play an essential role in the recovery of corals. Corals exposed to identical temperatures with a 50% reduction in irradiance had a 17% lower mortality rate than those at full light exposure.

• Species differences: No statistical differences in partial mortality or survivorship were found between heated and ambient conditions for L. scutaria. Significant differences occurred in calcification, bleaching, survivorship, and partial mortality for the other three species tested.

• Ambient temperature increase: Ambient mesocosm temperatures have increased by 2.3 °C between 1970 and 2017 with an offshore SST increase in the main Hawaiian Islands of 1.13 °C in the past 47 years.

• Improved water quality in Kāne‘ohe Bay: Inorganic nutrient levels have decreased dramatically since the removal of treated sewage disposal from the south bay in 1977–78. Available evidence indicates that the lower concentrations of nutrient pollutants, particularly dissolve organic nitrogen, have played an important role in the increased temperature tolerance of corals after nearly 50 years as was determined by these experiments.

Supplemental Information

Data S1 Raw data

Click here for additional data file.

Figure S1 Visual assessment of corals

Visual assessment of tested coral species throughout the stress (11 July–11 Aug 2017) and recovery periods. Scale (1, normal; 2, pale; 3, bleached; 4, dead).

Click here for additional data file.

Table S1 Mortality analysis

Cox proportional hazards regression analysis, with censoring of individuals that survived to the end of the experiment. Coral time to mortality was recorded as the number of days since the start of the experiment within each year.

Click here for additional data file.

We would like to thank our reviewers for their comments, which have improved the manuscript tremendously. We also thank D Coffey for his assistance in statistical analyses. This work is dedicated to Dr. Paul Jokiel, who defined the coral bleaching tolerances for Hawaiian corals. Without him, this work would not be possible. This publication is referenced as the University of Hawai‘i’s School of Ocean and Earth Sciences (SOEST) contribution number 10367 and Hawai‘i Institute of Marine Biology (HIMB) contribution number 1729.

Additional Information and Declarations

Competing Interests

Author Contributions

Field Study Permissions

Data Availability

The authors declare there are no competing interests.

Steve L. Coles conceived and designed the experiments, performed the experiments, authored or reviewed drafts of the paper, approved the final draft.

Keisha D. Bahr conceived and designed the experiments, analyzed the data, contributed reagents/materials/analysis tools, prepared figures and/or tables, authored or reviewed drafts of the paper, approved the final draft.

Ku’ulei S. Rodgers conceived and designed the experiments, contributed reagents/materials/analysis tools, authored or reviewed drafts of the paper, approved the final draft.

Stacie L. May, Ashley E. McGowan, Anita Tsang, Josh Bumgarner and Ji Hoon Han performed the experiments, approved the final draft, maintenance of experiments.

The following information was supplied relating to field study approvals (i.e., approving body and any reference numbers):

Corals were collected under HIMB Special Activity Permit 2018-03.

The following information was supplied regarding data availability:

The raw data are provided in a Supplemental File.

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
