# Peer review of "Evidence of acclimatization or adaptation in Hawaiian corals to higher ocean temperatures"

_PeerJ, doi:10.7717/peerj.5347_

## Round 0.1 · original submission · Major Revisions

Both reviewers agreed that this paper will make a strong contribution to this field and both were excited by the unusual ability to replicate a study from fifty years ago. While most comments related to some suggested wording or interpretation changes, the main concern from reviewer 2 was the lack of statistical comparison across time. I agree this could strengthen the rigor and conclusions for the study and is the basis for the "major revisions" decision. In approaching that analysis, take care to differentiate as much as possible between the actual treatments between the two times (actual temperatures), rather than the two times themselves (1997 vs. 2017) because the experiments were necessarily different. However, it really makes it hard to keep track of what is actually being compared.

As such, I suggest it will be necessary to more carefully lay out how the treatments from the two studies compare because it was diffiult for me to follow the actual treatments (especially since "ambient" and "elevated" meant two different actual temperatures across time). For instance, it wasn't obvious to me why there were four temperature treatments in 1970 compared to 2017 (as shown in Fig. 4). And it wasn't until I got to Fig. 6 that I really understood the actual temperatures that were being compared. Can Table 1 be expanded to incude the characters in the the 30 day stress period from the 1970s?

Revisiting how you display and analyze your results with a focus on comparisons between 1970s and 2017 I think will make these results more clear and powerful.

·

Basic reporting

The manuscript is largely well-written; suggestions for specific edits are in the general comments section below.

No raw data has been provided.

IACUC permit listed for fish used to control tank parasites, but no information on coral collection permits is listed.

Experimental design

The primary research question is timely and the methods used are sound. I have no concerns with respect to the methods used or statistical analyses conducted; however, the raw data has not been provided.

Validity of the findings

The discussion is to the point and succinct, but there are a few potential alternate explanations that I think would be worth discussing. First, the absolute tolerance of the selected coral species to the 31C heat stress increased across years, but the relative magnitude of the heat stress is not considered. In 1970, corals were exposed to a roughly +4.5C temperature increase, while in the present manuscript, the exposure was only a +2.5C change. The calcification data from 1970 (Fig. 4) suggests that a +2.5C change was also not as detrimental to coral performance, and I wonder how the survivorship curves compare at these temperatures to the 2017 experiment if that data is available? And whether a similar +4.5C increase in the present day would result in similar mortality rates. It would be unrealistic to collect additional data from a 34C stress treatment, but I think some discussion of the difference between relative and absolute thermal stress is worth including, and if the survival/bleaching data for the +2.5C treatments in 1970 are available, these would be worth including in the present manuscript as well.

In addition, the authors point out that other environmental variables have changed in Kaneohe Bay since the original experiment was run – namely, that water quality was also a major issue in 1970 – corals experienced higher nutrients, lower water clarity and decreased light levels. They hypothesize that this would have countered bleaching in the original 1970 experiments, but recent work on the role of symbiont density in driving coral bleaching suggests the pattern may be exactly the opposite - e.g. (Cunning and Baker 2014, Baker et al. 2018). Eutrophication favors the proliferation of symbionts (Fagoonee et al. 1999) and corals with greater symbiont densities are more susceptible to bleaching (Cunning and Baker 2013). This transition was also recently shown to be associated with an increase in symbiont parasitism – in which symbionts sequestered more resources for their own growth at a cost to the host (Baker et al. 2018). Consequently, is it possible that the difference in bleaching thresholds and growth deficits among years can be partially explained by a decrease in symbiont density across years as a result of improved water quality? Is there any way to get data on symbiont density? In any case, I think the potential role of changing water quality and possibly symbiont density should be considered in more detail.

Baker, D. M., C. J. Freeman, J. C. Y. Wong, M. L. Fogel, and N. Knowlton. 2018. Climate change promotes parasitism in a coral symbiosis. The ISME Journal 12:921–930.
Cunning, R., and A. C. Baker. 2013. Excess algal symbionts increase the susceptibility of reef corals to bleaching. Nature Climate Change 3:259–262.
Cunning, R., and A. C. Baker. 2014. Not just who, but how many: the importance of partner abundance in reef coral symbioses. Frontiers in Microbiology 5.
Fagoonee, I., H. B. Wilson, M. P. Hassell, and J. R. Turner. 1999. The Dynamics of Zooxanthellae Populations: A Long-Term Study in the Field. Science 283:843–845.

Additional comments

Though we are not supposed to comment on the “impact and novelty” of manuscripts, I nevertheless think this work will be of great interest to the coral research community. When asking about coral adaptation and/or acclimatization, we contemporary coral biologists substitute space for time – and compare corals inhabiting different reef localities. Re-running a nearly 50 year old experiment to ask how coral thermal tolerance has changed in time is hugely exciting!

My main criticisms are listed above in the ‘Validity of the findings’ section – my primary concern is that there may be alternate explanations for the difference in coral performance between the two experiments that are not adequately considered or discussed. Additional minor editing suggestions are listed below.

L20-21: I don’t think this sentence is needed and I think the abstract would be stronger without it.

L24-25: Can the bleaching result be worked into the prior sentence? This sentence is a bit confusing as written

L30-31: Given that irradiance did not alter calcification and only impacted mortality somewhat, I think this wording needs to be toned down or rephrased a bit.

L31-32: “the first strong evidence” is a bit vague – I suggest rewording to be more specific.

L51-55: This sentence is confusing, and moreover, does not reference the primary literature. I suggest rewording for clarity; and both here and in L56-58 finding more appropriate references.

L111-112: A reference is needed for this sentence.

L164: Information on coral collection permits is needed.

L344-345: The mean temperature increased, but was there also a concomitant change in variability? Did the range of temperatures experienced in a given day increase, decrease or stay the same?

·

Basic reporting

Overall, the writing is clear, professional, and well constructed. I note a few minor suggestions for clarification of the introduction and alternative citations. The introduction could be re-focused to target the relevant background for evidence for climate-change acclimatization/adaptation to make it stand out from the myriad of "corals are dying from climate change" introductions of standard bleaching papers. I also strongly suggest including raw data at the level of individual for all response variables. As presented, the data are not able to easily be re-analyzed.

Experimental design

The experimental design is sound and well-described. One question is why there were different numbers of individuals in the shaded and unshaded portions of each tank? This should be clarified and described specifically in the text as it creates an imbalance in sample numbers across treatments.

Validity of the findings

The authors present a very exciting and robust dataset, however there is one gap in the analysis that is serious enough to preclude a recommendation for acceptance at this time. I strongly suggest statistically analyzing the 1970 vs. 2017 comparisons directly for all of the data (Calcification, Survivorship, and Visual Assessment). Survivorship data could be analyzed as in "Oliver and Palumbi 2011 CoralReefs Do fluctuating temperature environments elevate coral thermal tolerance" with a cox hazard regression approach including study/year as a term in the model to directly compare resulting curves from 1970 and the 2017 experiments. Visual assessment data could be treated similarly to Oliver and Palumbi as well via some sort of trend-line fitting and statistical comparison of slopes. Calcification data could be compared via an ANOVA with study/year as a fixed factor. As it currently stands, all conclusions regarding differences between the years are based only on trends and not on any analyses. I don't expect the results to change, but this needs to be addressed prior to me recommending acceptance for publication.

Additional comments

General line-by-line comments below:
line 40-41, suggest modifying the area to the actual statistic of 0.17% of the world ocean area (sensu Smith 1978 Nature Coral-reef area and the contributions of reefs to processes and resources of the world's oceans) and updating the diversity stat to 33% of marine animal species (sensu Fisher et al 2015 CurrBiol Species Richness on Coral Reefs and the Pursuit of Convergent Global Estimates). The McAlister reference is difficult to access and I haven't been able to validate it's use in this instance while Smith 1978 presents an exact calculation and Fisher 2015 is a more updated diversity reference.
line 43, the phrasing "once climate change exceeds 1-2°C above the preindustrial" is confusing. Suggest clarifying to "preindustrial global/ocean temperatures" or similar specific term.
line 46, actually coral bleaching is first attributed to Vaughan 1914 (see first sentence of Jokiel and Coles 1990). Vaughn reference can be accessed via archive.org (or email me and I can send you a copy).
lines 52-53, again please be specific with your temperatures, are these ocean/sea-surface/atmospheric/etc. temperatures?
lines 55, but see (Xu et al 2018 JGeophysRes Evidence for the Thermal Bleaching of Porites Corals From 4.0 ka B.P. in the Northern South China Sea)
line 59, maybe cite Hughes 2017 Nature?
line 89 and 92, suggest citing Liu et al 2014 RemoteSense Reef-Scale Thermal Stress Monitoring of Coral Ecosystems New 5-km Global Products from NOAA Coral Reef Watch or Liu,G.,Rauenzahn,J.L.;Heron,S.F.;Eakin,C.M.;Skirving,W.J.;Christensen,T.R.L.;StrongA.E.;
Li, J. NOAA Coral Reef Watch 50 km Satellite Sea Surface Temperature-Based Decision Support System for Coral Bleaching Management; NOAA Technical Report NESDIS 143; NOAA/NESDIS: College Park, MD, USA, 2013; p. 33. over the website.
lines 109 and 111, suggest citing references for these definitions.
lines 134-135, add Jokiel and Coles 1990 as well
line 139, but what about Logan et al 2014 GCCBiol Incorporating adaptive responses into future projections of coral bleaching?
line 163, re-format citation to match others
line 182-187, citations for the breeding modes?
line 192, re-format last citation to match others (as well as others throughout listing full authors)
lines 314-322, why are there sometimes two different p-values for the same comparisons? e.g., the p<0.0001 and p=0.0006 on lines 321-322?
lines 356-360, were survivorship differences tested statistically? Could do a chi-squared to compare percent survival at the end or a repeated-measures ANOVA to compare survivorship percentages across time with fixed effects of temperature, day, and individual as the repeated measure.
line 384, use past tense "is" to "was" for consistency.
line 387-388, this is the second time the authors use a similar statement referenced to Coles and Brown 2003, though the statement is a bit misleading. I believe they are referring to naturally occurring, long-term differences in thermal thresholds between similar corals in different geographic locations (i.e., thermal environments). However, here, they are clearly demonstrating short-term shifts in thermal tolerances within the past 46 years due to warming seas. I think it is safe to say that this is the first time that thermal acclimatization/adaptation to anthropogenic warming has been experimentally determined in corals, a very different phenomenon from the naturally occurring geographic variation across latitudes.
lines 392-411, based on my reading of the results, none of these statements are supported by any statistical comparisons. The only statistical comparison between 1970 and 2017 results was the regression analyses for the temperature data.

---

## Round 0.2 · Minor Revisions

As you prepare your manuscript for final submission - please take note of the suggestions of the reviewer. Also, continue to work to make it as accessible as possible for non-specialist. Some final suggestions to increase accessibility:

1) I think it would benefit the manuscript to clarify the distinction between acclimitization and adaptation even more clearly. Although you don't differentiate here - what would it take to distinguish these two dynamics?

2) Continue to clarify the differences between the two expeiments - especially clarifying the different temperature treatments early on. It doesn't really become clear until you see the later figures that include all the treatments on the graph.

Overall though, a very interesting and unique submission.

·

Basic reporting

The revisions adequately addressed my concerns.

Experimental design

The revisions adequately addressed my concerns

Validity of the findings

The revisions adequately addressed my concerns

Additional comments

A few minor comments below. This should not require re-review:

Lines 47-49, it's actually not super clear who first described bleaching. See below summary and references from Glynn 1996. It sounds like maybe the Vaughn 1911 (not 1914) though I haven't found a copy of this to verify:

The first documented account of bleaching was reported by Goreau (1964), off the south coast of Jamaica in 1963. Two earlier reports that are cited to indicate that bleaching is not a recent phenomenon are Vaughan (1911) and Yonge & NichoIIs (1931). Vaughan (1911) used the term 'bleached' to describe the condition of the upper surfaces of massive corals that were periodically exposed during extreme, midday low tides in the Florida Keys. According to this report the stress (subaerial exposure) and response were presumably acute, causing rapid tissue death and sloughing. This resulted in a bare white skeleton that was soon invaded by epibenthic fouling species. This is not coral bleaching. The bleaching described by Yonge & NichoIIs (1931) did not occur naturally, but was induced by experimental treatments. Natural bleaching events must certainly have occurred in response to stress condi- tions in the past, but it is intriguing that they are so rarely mentioned until the 1970s. This suggests that the increased incidences of coral reef bleaching observed toward the end of this century are genuine. However, the absence of records of earlier periods of bleaching (from 1870 to 1970) are enigmatic, perhaps a result of non-occurrence, but also possibly due to non-reporting.
Goreau TF (1964) Mass expulsion of zooxanthellae from Jamaican reef communities after Hurricane Flora. Science, 145, 383-386
Vaughan TW (1911) The Recent Madreporaria of southem Florida. Carnegie Institution of Washington Yearbook (1910), 9, 135-144.
Yonge CM, Nicholls AG (1931) Studies on the physiology of corals. V. The effects of starvation in light and in darkness on the relationship between corals and zooxanthellae. Scientific Reports of the Great Barrier Reef Expedition 7928-29, 1, 177-211

Line 99, suggest changing to "globally,"?

Lines 172-173, Repeated from earlier in the paragraph

Line 405 "has" not "have"?

Line 486 "finding[s]"

Line 503 Cite Bruno papers here?

Conclusions (is the bulleted format allowed?)

---

## Round 0.3 · accepted · Accept

Congrats on a great paper!

#